# The Role of the Environment in Dynamics of Antibiotic Resistance in Humans and Animals: A Modelling Study

**DOI:** 10.3390/antibiotics11101361

**Published:** 2022-10-05

**Authors:** Hannah C. Lepper, Mark E. J. Woolhouse, Bram A. D. van Bunnik

**Affiliations:** 1Usher Institute, Ashworth Laboratories, University of Edinburgh, Edinburgh EH9 3FL, UK; 2Roslin Institute, University of Edinburgh, Edinburgh EH25 9RG, UK

**Keywords:** antibiotic resistance, One Health, mathematical model, AMR in the environment

## Abstract

Antibiotic resistance is transmitted between animals and humans either directly or indirectly, through transmission via the environment. However, little is known about the contribution of the environment to resistance epidemiology. Here, we use a mathematical model to study the effect of the environment on human resistance levels and the impact of interventions to reduce antibiotic consumption in animals. We developed a model of resistance transmission with human, animal, and environmental compartments. We compared the model outcomes under different transmission scenarios, conducted a sensitivity analysis, and investigated the impacts of curtailing antibiotic usage in animals. Human resistance levels were most sensitive to parameters associated with the human compartment (rate of loss of resistance from humans) and with the environmental compartment (rate of loss of environmental resistance and rate of environment-to-human transmission). Increasing environmental transmission could lead to increased or reduced impact of curtailing antibiotic consumption in animals on resistance in humans. We highlight that environment–human sharing of resistance can influence the epidemiology of resistant bacterial infections in humans and reduce the impact of interventions that curtail antibiotic consumption in animals. More data on resistance in the environment and frequency of human–environment transmission is crucial to understanding antibiotic resistance dynamics.

## 1. Introduction

Antibiotic resistance (AMR) is a One Health issue, with bacterial species carrying resistance genes, such as *E. coli*, *Salmonella*, and *Campylobacter*, being able to colonise and transmit between humans, animals, and the environment [1,2,3]. Sharing of resistant bacteria between humans and animals due to zoonosis has been observed (e.g., [4]) and modelled (e.g., [5]). Now, the potential of the environment for dissemination of AMR is being increasingly recognised, for example, as a result of the volume of resistance bacteria in human and agricultural wastewater effluent being discharged into natural waters and soils [6,7,8].

There are many potential routes for resistant bacteria into the environment. Several studies have demonstrated the likelihood that resistant bacteria in humans can be transferred to the environment, including rivers [9], coastal waters [10], and soils [11]. In addition, studies have linked resistant bacteria in animals with their respective environments, such as between wild animals and human-impacted environments [12,13], as well as between livestock and their environment, especially in aquaculture [14,15]. However, the risk that resistance in the environment poses to humans and animals remains poorly understood [16]. 

Mathematical models are an important tool to study complex dynamics inherent in the emergence and spread of resistance [17] and can therefore be used to improve our understanding and combat the spread of AMR in humans, animals, and the environment. However, a lack of data and understanding about the burden, selection, and transmission of resistant bacteria, especially in animals and the environment, presents a challenge with respect to parameterising models of AMR from a One Health perspective. Consequently, there are few models of resistant bacteria that connect humans, animals, and the environment [18]. 

Some existing studies incorporate an environmental component into transmission models of resistant bacteria in hospitals or farms. Two compartmental models demonstrated that reducing or eradicating resistance in a hospital setting was more difficult to achieve when the environment was explicitly modelled [19,20]. Studies taking the environment into account when modelling the spread of resistance in farms have revealed that environmental parameters were key in dynamics of resistance in the farm [14,21]. However, a recent modelling study revealed that interventions to reduce antibiotic consumption in animals would still be effective when the influence of resistance in animals and the environment is considered [22]. These findings indicate the need for further exploration of the role of the environment with fully dynamic transmission models.

In this study, we aimed to investigate the importance of the environment on the long-term dynamics of resistant bacterial infections in humans, including how it might affect the impact of interventions to reduce resistance in humans. We developed a compartmental model of resistance transmission within and between humans, animals, and the environment. We used a dynamic environmental compartment, improving on existing models, which allowed us to assess the importance of within-environment processes. Our objectives were: (1) to investigate how adding an environmental compartment affects the long-term dynamics of resistance in humans, as well as the sensitivity of the model to its parameters; and (2) to investigate the impact of interventions to curtail antibiotic usage in animals or the environment to human transmission on the prevalence of resistance in humans in this model. 

## 2. Results

All analyses were conducted in both bounded and unbounded environmental capacity versions of the model.

### 2.1. Long-Term Dynamics of Resistance in Humans

#### 2.1.1. Prevalence of Resistance in Humans

Parameter sets were identified that corresponded to the intended target equilibrium human resistance prevalence of 54% in all transmission scenarios and in both the bounded and unbounded versions of the model (Appendix A) (the resistance rate to aminopenicillin in blood and cerebrospinal fluid sample isolates of *E. coli* in the EU in 2020 [23]). Figure 1B shows that the amount of resistance in the environment was influenced by the model structure and the transmission scenario. The highest level of resistance in the environment was 0.23 in the environment-driven, unbounded version of the model, indicating that an implausibly high level of environmental contamination is not needed for observed human resistance levels. 

#### 2.1.2. Sensitivity Analysis

Model sensitivity results are presented in Figure 1C. In both bounded and unbounded models, human resistance prevalence was most sensitive to $\mu_{H}$, the rate of loss of resistance from humans, but relatively insensitive to $\Lambda_{A}$, the antibiotic consumption in animals. The rate of transmission from the environment to humans ($\beta_{EH}$) was at least as important as $\beta_{HH}$ and $\beta_{AH}$, the rates of transmission to humans from other humans and from animals, respectively. Moreover, $\beta_{EH}$ is more influential than any other transmission parameter in the unbounded model. The rate of loss of resistance from the environment ($\mu_{E}$) was more important for human resistance levels in the unbounded than the bounded model.

### 2.2. Impact of Interventions to Reduce Resistance in Humans

#### 2.2.1. Impact of Curtailing Antibiotic Usage in Animals

Curtailing antibiotic usage in animals had a small impact on human resistance levels when the animals contributed less to resistance transmission (Figure 2). The percentage decrease in human resistance levels achieved by this intervention with or without an environmental compartment was similar, indicating that sharing transmission between the three compartments as opposed to the two compartments did not affect the effectiveness of this intervention. The animal-driven transmission scenario had the highest impact (22% decrease in human prevalence), and the human-driven scenario had the lowest impact (0.1%). In the environment-driven transmission scenario, the environmental capacity was influential; when bounded, the impact was low (0.6%) and increased marginally when unbounded (0.7%). Both the environmental structure and the transmission parameters affected the impact of antibiotic usage reduction in animals.

#### 2.2.2. Reducing $\Lambda_{A}$ vs. Reducing $\beta_{\mathrm{EH}}$

We compared the impact (ω) of reducing either $\Lambda_{A}$ (antibiotic consumption in animals) or $\beta_{EH}$ (transmission of resistant material from the environment to humans) (Figure 3). We considered pre-intervention values of 0.1 for each parameter, as well as the impacts in different transmission scenarios. This value was chosen so that the size of the intervention was consistent between transmission scenarios. Interventions to reduce $\beta_{EH}$ had a greater impact than interventions to curtail $\Lambda_{A}$ when transmission was human- or environment-driven. Curtailing $\Lambda_{A}$ had a greater impact when animals dominated transmission or when transmission was balanced between the compartments. .

#### 2.2.3. Effect of $\beta_{EH}$ on Impact of Interventions to Reduce Antibiotic Consumption in Animals

We next identified the impact of reducing $\Lambda_{A}$ across a range of values for $\beta_{EH}$ (Figure 4). In the balanced and environment-dominated transmission scenarios, increasing $\beta_{EH}$ initially increased and then decreased the size of the impact of curtailing antibiotic usage in animals in all transmission scenarios (Figure 4). This peaked shape of the impact size is caused by the increase in $\beta_{EH}$, allowing a greater amount of indirect (or environment-mediated) transmission in animals and humans. Initially, the increased animal–human indirect transmission means that human resistance levels are more sensitive to resistance dynamics in animals, leading to an increase in impact for the intervention. However, as the level of indirect transmission continues to increase, the environmental reservoir becomes more influential, and human resistance levels become less sensitive to changes in $\Lambda_{A}$. In the animal-dominated transmission scenario, on the other hand, because the resistance dynamics in animals are more influential than the environment over a wider range of $\beta_{EH}$. values, the impact of reducing antibiotic consumption in animals continues to increase with increased $\beta_{EH}$. Appendix A shows that an increase in intervention impact was also observed across the range of pre-intervention values for $\Lambda_{A}$. These results indicate that increasing environmental transmission could improve or reduce the impact of curtailing antibiotic usage in animals.

## 3. Materials and Methods

### 3.1. Model Description

We extended the original model presented by van Bunnik and Woolhouse (2017) [24] to include an environmental compartment. Humans and animals gain resistant infection by exposure to antibiotics or exposure to other humans, animals, or environments carrying resistant bacteria. Resistance in the environmental compartment is increased by contact with humans or animals who carry resistant bacteria or via exposure to antibiotics that have been excreted by humans or animals. The environment compartment does not consider any particular type of environment, such as water or soil, but a summation of these types.

We defined the model using a system of coupled ordinary differential equations:(1)$$\begin{array}{c} \frac{dR_{H}}{dt}=\left(1-R_{H}\right)\cdot \left(\Lambda_{H}+\beta_{HH}\cdot R_{H}+\beta_{AH}\cdot R_{A}+\beta_{EH}\cdot R_{E}\;\right)-\mu_{H}\cdot R_{H}\; \end{array}$$
(2)$$\frac{dR_{A}}{dt}=\left(1-R_{A}\right)\cdot \left(\Lambda_{A}+\beta_{AA}\cdot R_{A}+\beta_{HA}\cdot R_{H}+\beta_{EA}\cdot R_{E}\right)-\mu_{A}\cdot R_{A}$$
(3)$$\begin{array}{c} \frac{dR_{E}}{dt}=\gamma_{H}\Lambda_{H}+\gamma_{A}\Lambda_{A}+\beta_{HE}\cdot R_{H}+\;\beta_{AE}\cdot R_{A}-\mu_{E}\cdot R_{E} \end{array}$$
where $R_{H}$ and $R_{A}$ are the fractions of the human and animal population that are infected with resistant bacteria, respectively; $R_{E}$ is a measure of the amount of resistant infectious bacteria in the environment; $\Lambda_{H}$ is the constant rate at which resistance is gained from exposure to antibiotics in humans; and $\Lambda_{A}$ is the equivalent in animals. These are composite variables, taking into account both the amount of antibiotics consumed and the rate at which selection causes resistance in bacteria to arise. $\mu_{H}$ is the reversion rate of humans infected with resistant bacteria to having only sensitive bacteria, and $\mu_{A}$ is the equivalent in animals. This includes the rate of clearance of resistant infection and the rate of death in a fixed-size population. The parameters $\gamma_{H}$ and $\gamma_{A}$ are scaling parameters determining how much of antibiotic exposure in humans ($\Lambda_{H}$) and animals ($\Lambda_{A}$) will result in excreted antibiotics, selecting for an increase in resistant bacteria in the environment. $\mu_{E}$ is the rate of loss of resistant infectious bacteria from the environment. Transmission within and between the compartments is controlled by the β transmission coefficients, with the subscripts indicating the direction of transmission of each coefficient. For example, $\beta_{HH}$ is the transmission coefficient between humans, and $\beta_{EH}$ is the transmission from the environment to humans.

Further details about parameter definitions, units, and value ranges can be found in the Appendix A. Figure 1A shows a flow diagram representing the movement of infectious resistant material between and within the different compartments. All rates are per capita with respect to the human and animal populations and per environmental unit with respect to the environment (see next section). We used the numerically obtained steady-state solutions of this model, as we were interested in long-term prevalence. The timestep of the model represents one month. Determination of the time step is discussed in the Appendix A (Additional Methods Information).

### 3.2. Capacity for Resistance in the Environment

Equation (3) represents the environment as an unbounded compartment in which the amount of resistant infectious material in the environment is in the range of 0–∞. We consider one “unit” of the environment to be the human infectious potential equivalent. This means that for a value of $R_{E}=1$, if the transmission coefficients $\beta_{EH}$ and $\beta_{HH}$ were the same, i.e., each unit of the environment would transfer resistant material to humans at the same rate that an infected human would transfer resistant material to another human. Although theoretically, the environment may have a maximum capacity for resistant material, we do not have a way to determine this capacity, so we modelled the environment as an unbounded compartment. For comparison, we also explored a version of the model in which resistance levels in the environment cannot exceed 1. In this model, the environmental compartment is specified as :(4)$$\begin{array}{c} \frac{dR_{E}}{dt}=\left(1-R_{E}\right)\cdot \left(\gamma_{H}\Lambda_{H}+\gamma_{A}\Lambda_{A}+\beta_{HE}\cdot R_{H}+\;\beta_{AE}\cdot R_{A}\right)-\mu_{E}\cdot R_{E}\; \end{array}$$

This model assumes that there is no growth or dissemination of resistant organisms within the environment. We also assume that the environment is only exposed to antibiotics that are excreted by humans or animals. The environment may be exposed to antibiotics directly through, for example, the effluent of pharmaceutical industries, but we do not consider this specific case here.

### 3.3. Impact of Interventions on Resistance in Humans

We investigated the impact of two types of interventions on the levels of resistance in the human compartment. First, we considered interventions to reducing antibiotic usage in livestock (reducing $\Lambda_{A}$ to zero) and how changes to environmental parameters affect the effectiveness of this intervention. Secondly, we considered interventions that would reduce the transmission of resistant bacteria from the environment to humans (reducing $\beta_{EH}$ to zero).

We measured the impact of interventions as the percentage decrease in resistance levels in humans according to the procedure described by van Bunnik and Woolhouse (2017). We compared equilibrium values of $R_{H}$ before ($R_{H}^{*}$) and after the intervention ($RI_{H}^{*}$), to obtain the impact or percentage decrease in human resistance levels:(5)$$\omega =1-\frac{RI_{H}^{*}}{R_{H}^{*}}$$

We investigate the impact of reducing $\beta_{EH}$ and that of curtailing antibiotic usage in animals ($\Lambda_{A}$).

### 3.4. Sensitivity Analysis

We used the extended version of the Fourier amplitude sensitivity test (FAST) [25] to analyse the relative influence of each parameter on the value of $R_{H}$, the outcome measure of interest. A total sensitivity index for each parameter was calculated based on the variance of $R_{H}$ over the variation in all parameters. The R package fast was used for this analysis [26].

### 3.5. Parameterisation

To the best of our knowledge, there are no direct estimates for the rate of emergence of resistance due to antibiotic exposure ($\Lambda_{A}$ and $\Lambda_{H}$), nor for rate of emergence of resistance in the environment due to exposure to antibiotics derived from animals ($\gamma_{A}$ and $\gamma_{H}$). Although there are promising data available for estimation of some transmission rates [27,28], these are currently limited, providing information only on the number of possible transmission events between humans and animals, without environmental sampling. An estimate of the number of transmission events between humans and animals does not indicate the direction of transmission (i.e., human to animal or animal to human), nor the rate of transmission, as no time period for the observed events can be provided, nor how much of this transmission was environmentally mediated. Therefore, all transmission parameters referenced in this study are estimated through ordinary least squares minimization (e.g., [29]).

However, some estimates may be obtainable with respect to the rate of loss of resistance in humans, animals, or the environment. The duration of infection in humans can be approximated using studies that take longitudinal samples from patients and that have a good estimate of time of infection. One study of transient colonisation with resistant E. coli following international travel found 83% of colonisations were cleared after 6 months (a clearance rate of 0.2 per month) [30], whereas another study revealed that 85% of colonisations due to a resistant E. coli infection were cleared after 3 years (a clearance rate of 0.03 per month) [31]. We chose the mean of these two rates for our estimate of clearance rates in humans (0.118). Finding a similar statistic for livestock is challenging. Some studies considered repeated longitudinal samples from young livestock (e.g., around the time of weaning) but did not identify the time of infection [32,33]. Other studies identified the time of infection but used a virulent strain of E. coli, which may not be representative of the general clearance rate of non-pathogenic, resistant E. coli [34]. However, these studies do suggest that the duration of infection in pigs and cattle is short, e.g., cleared in weeks or months [32,33,34]. Therefore, we doubled the value for humans as an estimate of the clearance rate in livestock (0.24). An estimate of the rate of loss of resistant bacteria through degradation in the environment of 0.29 per month was obtained from an experimental study of E. coli in decomposing cattle manure [35].

Due to a paucity of data about many of the parameters included in the model, we aimed to explore a wide range of parameter scenarios in this model. We chose the following transmission scenarios: (1) a balanced transmission scenario, with all transmission coefficients equal; (2) human-driven transmission (i.e., if the subscript H denotes humans and x denotes any other compartment, $\beta_{Hx}>\beta_{xx}$); (3) animal-driven ($\beta_{Ax}>\beta_{xx}$); and (4) environment-driven ($\beta_{Ex}>\beta_{xx}$). Non-identifiability between transmission parameters means that selecting one set based on fitting to human prevalence data may not be reliable. For example, with no other information about transmission rates, a given prevalence in humans may be equally likely to be caused by high human-to-human transmission, even if other transmission rates are low, or by high animal- or environment-to-human transmission. We also used multiple transmission scenarios to explore ways of generating the observed human resistance prevalence.

We also averaged our results across parameter sets generated randomly using sampling distributions for the three parameters ($R_{H}$) that were most sensitive to (viz., $\mu_{H}$, $\mu_{E}$, and $\Lambda_{H}$), to avoid over-reliance on model dynamics that are unusual to a particular combination of parameters rather than generally true of the system. All parameter values and sampling distributions can be found in the Appendix A (Appendix A), as well as the methods for obtaining transmission scenario parameters.

### 3.6. Software

Analyses were carried out using Wolfram Mathematica version 11.3 [36], R 4.1 [37], and Julia 1.7 [38]. The code for the model, parameter set generation, and visualisations is available at https://github.com/hannahlepper/animal-human-env-model (created 29th January 2019). 

## 4. Discussion

### 4.1. Key Findings

In this study, we modelled the transmission of resistant bacteria between humans, livestock animals, and the environment and assessed the impact of interventions that reduce antibiotic consumption in animals or decrease the transmission of resistant bacteria from the environment to humans. We found that antibiotic resistance prevalence in humans is sensitive to transmission between humans and the environment. Interactions between the transmissibility of the environmental reservoir and the impact of curtailing antibiotic usage in animals are complex, with greater transmissibility increasing impact by linking animals and humans more closely in some cases and mitigating the impact in other cases. Reducing the transmission of resistant bacteria from the environment to humans was found to be a more effective intervention than reducing antibiotic consumption in animals if humans or the environment dominated transmission. Overall, these results indicate that resistant bacteria in the environment can influence the prevalence of resistance in humans. The magnitude of environmental influence will depend on the amount and dynamics of resistant bacteria in the environment. Assessing the likelihood of observing these theoretical results in the real world is hindered by a lack of quantified, generalisable data on the types, amount, and degradation of resistance in the environment, as well as the transmission of resistance between humans, livestock, and the environment.

### 4.2. Is Curtailing Antibiotic Usage in Animals an Effective Intervention to Reduce Human Resistance Levels?

The greatest observed impact of curtailing antibiotics in animals was a 22% decrease in the human resistance level in the animal-dominated transmission scenario, and the smallest impact was a 0.1% in the human-driven transmission scenario. This result does not theoretically support the notion that curtailment of antibiotics would appreciably decrease resistance in humans in all settings. In contrast, there is some empirical evidence that curtailing antibiotics in livestock could reduce human resistance levels, although from a small set of observational studies [39]. A study of use of third-generation cephalosporin ceftiofur in broiler rearing in Canada revealed that resistance in Salmonella and E. coli was decreased in clinical isolates by 20% and 40%, respectively, after ceftiofur use decreased [40]. Other studies have revealed that animals do not contribute more to resistance patterns in humans than other humans [41] and that livestock–human transmission events are rarer than human–human transmission events; therefore, it is likely that a human-dominated transmission scenario is most realistic [27,28]. Therefore, the real-world population-level effect is greater than our results would predict, which may represent an underestimate, especially with respect to the degree of sharing of resistance between humans and animals. More data-based parameterisation will be crucial to improve the accuracy of One Health resistance transmission models. In addition, we examined the change in prevalence resulting from decreasing antibiotic usage in animals to zero. However, it is unlikely that livestock would have no exposure to antibiotics whatsoever, as they will sometimes be treated for infections. Incorporating this residual exposure to antibiotics could have led to lower estimates of the impact of curtailing antibiotic usage in animals and humans.

The size of the effect of intervening to reduce antibiotic consumption in livestock varied by transmission scenario (balanced transmission or transmission driven by either humans, livestock, or the environment). Therefore, a key question in terms of assessing the accuracy and relevance of the resulting intervention effect sizes is, ‘how realistic are the transmission scenarios?’. Although transmission of resistance between humans and animals is of considerable concern, evidence that conclusively demonstrates a case of direct transmission is rare [42,43]. Accurately parameterising the relationship between resistance in humans and livestock is an ongoing area of research [28] that will be crucial for One Health modelling of resistance.

As we increased the transmission rate from the environment to humans, the effectiveness of antibiotic curtailment was initially increased and then decreased in the balanced and environment-dominated scenarios. This suggests that the environment can provide a ‘back door’ transmission route from animals to humans that can reduce the effectiveness of antibiotic curtailment by adding to overall animal–human transmission rates. Using a two-pronged approach by intervening to reduce environmental transmission at the same time could therefore improve the impact of the curtailment of antibiotic usage. However, the effect of environmental transmission on antibiotic curtailment effectiveness was always negligible in the human-dominated transmission scenario (Appendix A), again indicating the importance of transmission setting for this result. It remains unclear whether non-human-dominated transmission scenarios are realistic and therefore what the real-world size of this back-door effect might be. There is some evidence that microbiomes in humans, animals, and the environment become more shared as interactions become more frequent [44], suggesting that transmission scenarios in which humans do not dominate transmission (such as the balanced and baseline scenarios) are possible. Further work to quantify environmental resistance concentrations and transmission could improve the accuracy of outcome predictions of antibiotic usage interventions. As reducing antibiotic usage in livestock animals is potentially a costly intervention [45], it is important to ensure optimal implementation.

### 4.3. Could the Environment Be an Effective Alternative Intervention Target?

The rate of transfer of resistant bacteria from environment to humans ($\beta_{EH}$) is also a potentially effective intervention target. Human resistance prevalence levels were sensitive to $\beta_{EH}$ and $\mu_{E}$, i.e., the rate of loss of resistant bacteria from the environment (sensitivity analysis, Figure 1C), which suggests that interventions to reduce how much resistance humans gain from the environment would be effective. The impact of reducing $\beta_{EH}$ was more effective than antibiotic usage curtailment interventions, although the difference was small in the animal-dominated scenario (Figure 2A). Interventions that improve sanitation have been proposed to reduce the occurrence of transmission of resistance between humans and the environment in informal urban communities in LMICs, where there is frequent exposure to resistance bacteria in the environment [46,47]. Nadimpalli et al. (2020) focused particularly on the potential benefits of improved water and wastewater infrastructure to control and prevent AMR transmission but noted that few studies have investigated the impacts of sanitation interventions on AMR. In high-income countries, some studies have demonstrated transmission of resistant bacteria via the hospital environment, which can be mitigated by increased cleaning and replacement of contaminated environments, such as p-traps (e.g., [48,49]).

### 4.4. Should the Environment Be Included in AMR Models?

In this model, the environment played an important role in the long-term dynamics of antibiotic resistance levels in humans. Mechanistically, the environment acts as a reservoir for antibiotic resistance from humans and animals in this model structure. Therefore, parameters that provide more opportunity for transmission to humans were influential in human resistance levels, especially the rate of loss or level of persistence of resistant bacteria in the environment ($\mu_{E}$). Environmental parameters were also influential in terms of the size of impact of interventions, and we demonstrated that it may be an effective intervention target itself. Existing models that incorporate an environmental component have also highlighted the potentially strong role the environment could play in increasing resistance levels in humans and undermining interventions [19,20,21,22]. Most models include the environment as a constant rather than a dynamic compartment, with the exception of that proposed by Booton et al. (2021). As our reported results are similar to those of models with constant compartments, this may indicate that models incorporating the environment non-dynamically are sufficient to account for this additional source of resistant bacteria. On the other hand, the model proposed by Booton et al. (2021) assumes that transmission of resistance (including from the environment) is dependent on exposure to antibiotics and therefore that human antibiotic usage is the most influential parameter for human resistance, downplaying the role of the environment. This contrasting result points to a need for further models that compare the contribution of the environment under different model structures and assumptions. Incorporating the environment into models of AMR spread may be important in terms of understanding AMR prevalence and for evaluating intervention success.

### 4.5. Modelling the Environment Highlights Data Needs

The results of our model highlight some key data needs for understanding the importance of AMR in the environment for humans. There are two influential parameters in the model that are difficult to parameterise from existing data: the rate of transfer of AMR from the environment to humans and the rate of loss of resistance in the human population.

How frequently humans gain resistant bacteria after exposure to an environmental source is unknown. There is evidence that humans can be exposed to resistant bacteria in the environment. For example, one study estimated that the amount of third-generation cephalosporin-resistant E. coli ingested by humans during recreational water use in coastal regions in England and Wales poses a risk of infection [10]. However, it is not clear how often such exposure leads to infection or colonisation [50]. More research that demonstrates a close relationship and epidemiological link between resistant bacteria colonising the environment and humans is needed to understand the frequency of environment–human transmission events. Use of high-resolution typing, such as whole-genome sequencing of, for example, isolates from hospital patients and the hospital environment in longitudinal studies, would be ideal for such research.

Studies have provided data on the rate of clearance of resistant infections in humans. A systematic review on methicillin-resistant S. aureus (MRSA) and vancomycin-resistant Enterococcus (VRE) colonisation revealed that it takes a period of 88 and 26 weeks, on average, to clear MRSA and VRE infections, respectively [51]. However, these studies note that there is considerable methodological heterogeneity in studies of MRSA and VRE, including varying definitions of clearance and length of follow-up [51]. The studies also focused primarily on hospital-associated resistance. Data on resistant bacteria colonisation prevalence and clearance in the community, where the role of exposure to animals and the environment may play a greater role, appear to be rare. Parameterising generalisable One Health models will therefore be benefitted by more research into resistance in the community.

### 4.6. Limitations

This study is subject to some important limitations that should be noted. First, we made simplifying assumptions in the structure and parameterisation of the model in response to the questions posed in this study; however, there are still many complexities with respect to the spread and emergence of AMR in humans, animals, and the environment to be explored. Future models should explore the importance of potential complexities, such as heterogeneity of transmission events, separate humans-specific and animal-specific environmental reservoirs, variation in the capacity for resistance in the environment, or the fitness costs to bacteria of carrying resistance in the three populations.

We did not model the dynamics of transmission of resistant bacteria and resistance genes separately but assumed that transmission parameters combine the transmission of both. This is in keeping with the assumptions of the original model [24]. Resistance genes can be transferred between bacteria via plasmid transfer or bacteriophages and can also be lost from bacterial lineages. The transmission rates of resistance genes in the human population may therefore differ from those of resistant bacteria, representing a limitation that was not captured in our model. AMR epidemiology and surveillance are usually measured in resistant bacteria, so there is little data on the prevalence and transmission rates of specific resistance genes.

Two further assumptions about resistance in the environment are that there is no growth of resistant material within the environment and that all antibiotics secreted into the environment are from human and livestock usage. The dynamics of resistance genes and bacteria in the environment is a complex topic, and although there are potentially environments in which resistance may spread (especially in sewage) additional empirical and modelling research is needed [50,52]. A recent review revealed that the sources of antibiotics in groundwater include excretion from humans and animals (via sewage and manure), as well as landfill, aquaculture, and industrial sites [53]; therefore, not including these sources may limit the accuracy of the results of this model. However, the relative contribution of each source is not well-known and may vary from one country to another [53].

## 5. Conclusions

This study illustrates the potentially important role of the environment in the epidemiology of resistant bacterial infections in humans. We highlight the need to consider the role of the environment in the design of AMR control strategies, as it can be influence human prevalence of resistance, reducing the effectiveness of interventions that curtail antibiotic consumption in animals, and may be an effective intervention target itself via improved sanitation infrastructure. Incorporating the environment into a One Health model of antibiotic resistance as a dynamic compartment was useful for considering the role of the environment. However, assessing the uncertainty of model predictions is hindered by a lack of data on the types and frequency of resistance in the environment, as well as the frequency of environment–human transmission events.

## Figures and Tables

**Figure 1 antibiotics-11-01361-f001:**
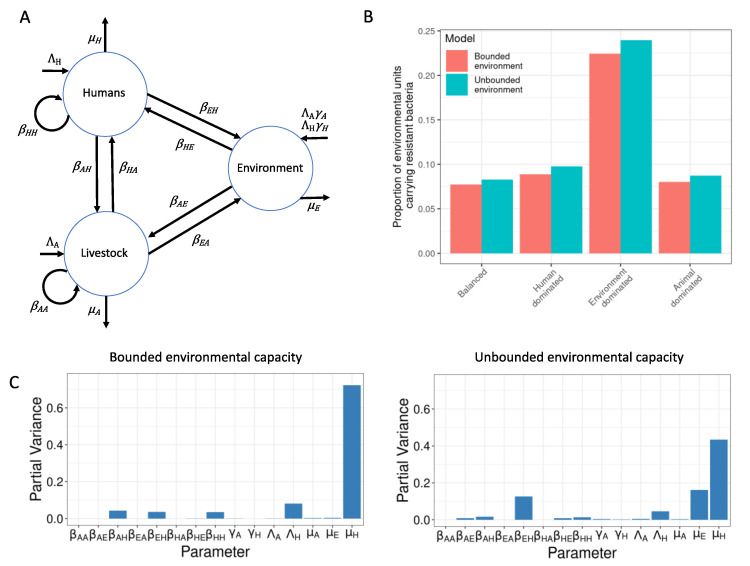
(**A**) Flow diagram indicating the model structure. (**B**) $R_{E}$ values in all transmission scenarios and both model structures. (**C**) Fourier amplitude sensitivity tests (FAST), indicating how much variation in $R_{H}$ was explained by each model parameter. On the left, FAST for the version of the model in which $R_{E}$ is bounded to 1. On the right, FAST for the version of the model in which $R_{E}$ was unbounded.

**Figure 2 antibiotics-11-01361-f002:**
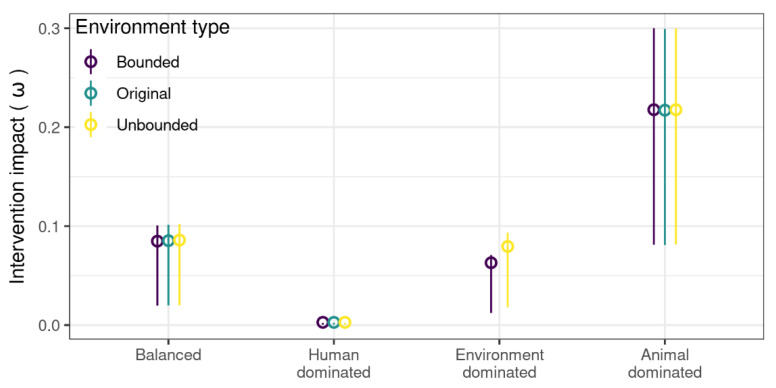
Mean impact of reducing $\Lambda_{A}$ from 0.1 to 0 across transmission scenarios. Transmission scenarios were specified so that transmission between humans was balanced or driven mainly by humans, the environment, or animals in the “balanced”, “human-”, “environment-”, and “animaldominated” scenarios, respectively. Results were averaged for parameter sets with varying values of $\mu_{H}$, $\mu_{E}$, and $\Lambda_{H}$. Error bars indicate 25% and 75% quantiles.

**Figure 3 antibiotics-11-01361-f003:**
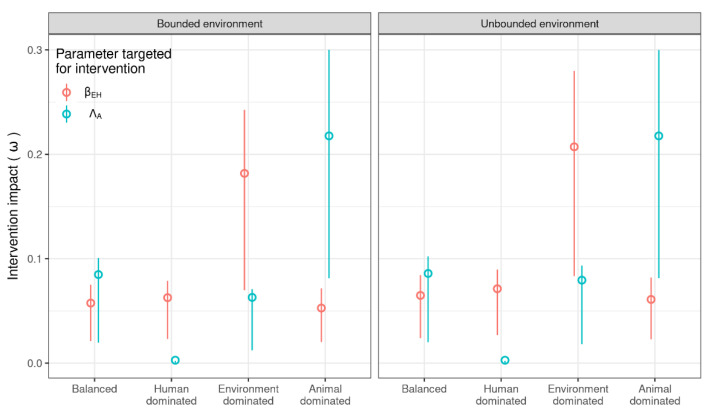
Impact (proportion decrease in $R_{H}$ after the intervention) of reducing either $\beta_{EH}$ or $\Lambda_{A}$ in all transmission scenarios and for both model structures. The intervention target was reduced from 0.1 to 0 in each case for consistency. Open circles indicate the average value of impact, and error bars show 25% and 75% quantiles.

**Figure 4 antibiotics-11-01361-f004:**
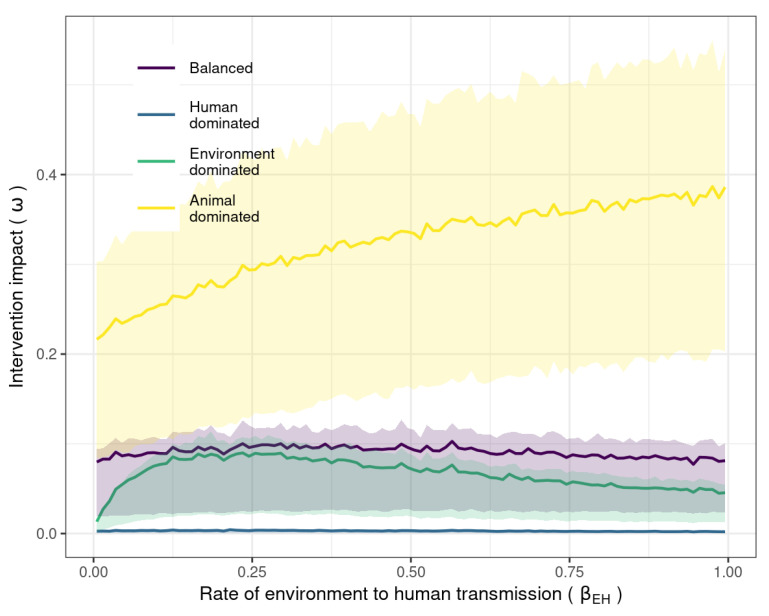
Mean impact of antibiotic decrease in animals on human resistance levels (proportion decrease in human resistance levels) for each transmission scenario with an increasing rate of environment-to-human transmission ($\beta_{EH}$). Ribbons indicate 25% and 75% impact quantiles.

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
