# Peer review of "The Role of the Environment in Dynamics of Antibiotic Resistance in Humans and Animals: A Modelling Study"

_antibiotics, 2022, doi:10.3390/antibiotics11101361_

Round 1

Author Response

Point 1: Regarding to manuscript antibiotics-1813534, entitled ” The role of the environment in transmission of antibiotic resistance between humans and animals: a modelling study”, Authors develop a model system for intervention to reduce antibiotic resistance. However, authors shall define this manuscript for antibiotic resistance or antibiotic resistant bacteria.

Response 1: We thank the reviewer for this comment. However it is not clear what is meant by the suggestion to “define this manuscript for antibiotic resistance or antibiotic resistant bacteria”.

Point 2: The following is a summary of transmission and development of antimicrobial resistance. Authors may be used to revise the manuscript, especially to use a zoonotic bacterial species as a model to elucidate the model.

  1. Antibiotic resistance in Animal farms
    1. Pre-existing antibiotic resistant bacteria (ARB)
      • 1. Farm conditions: Feeding, equipment, soil, wild animals, etc.
      • 2. Sanitation: Transfer from worker to animals
      • 3. Animal source a carrier

  1. Transmission from animals to human
    1. Transportation
  1. Slaughter process
  2. Marketing sale

  1. Antibiotic sensitive bacteria → ARB
    1. Misuse of antibiotics to induce mutations at
      • antibiotic targeting genes,
      • porin for antibiotic entrance
      • Expression of efflux pump, such as induction by insertion sequence
    2. transmission of antibiotic resistant genes
      • plasmid, integrin, transposon, insertion sequence, genomic island
    3. transformation

  1. Zoonotic bacteria
    • Food-poisoning bacteria
    • Limited species
    • Sometime related to immune-compromised conditions.

Response 2: We have selected a zoonotic bacterial species (E. coli) to study. Clearance rate parameters were found to suit E. coli, and the transmission parameters in each transmission scenario were fit to achieve resistance levels in E. coli. These methods are described in further detail in lines 181 – 197 and 231 – 233 in the revised manuscript.

In addition, we have clarified the language and added some references to the introduction to highlight specific examples of transmission of bacteria between humans, animals, and the environment (lines 45 – 51). We have added the following references: Collignon, P. J., & McEwen, S. A. (2019). One health-its importance in helping to better control antimicrobial resistance. Tropical Medicine and Infectious Disease, 4(1); Li, J., et al. (2019). Inter-host transmission of carbapenemase-producing Escherichia coli among humans and backyard animals. Environmental Health Perspectives, 127(10), 1–11; and Singer, R. S., et al (2007). Modeling the relationship between food animal health and human foodborne illness. Preventive Veterinary Medicine, 79(2–4), 186–203.

Reviewer 2 Report

Dear Authors

The article topic is very interesting because it meets the lack of information came from the environmental sector. However,  there are some major revisions, because it is very difficult to understand the results obtained from the mathematical model due to the lack of source of data input used. Furthermore no clear relevance between the title and the manuscript. 

Author Response

Point 1: The article topic is very interesting because it meets the lack of information came from the environmental sector. However, there are some major revisions, because it is very difficult to understand the results obtained from the mathematical model due to the lack of source of data input used.

Response 1: We thank the reviewer for these comments. We have added source data on clearance rates of E. coli from humans, livestock, and soils (described in revised manuscript on lines 181 – 197), and fit the transmission scenarios to the rate of resistance to aminopenicillins in E. coli isolates from patients in the EU in 2020 from the European Antimicrobial Resistance Surveillance Network (EARS-Net) (lines 231 – 233). Changes to the results arising from this change to the methods are reflected in the following parts of the text: the abstract (lines 23 – 24 and 28 – 29), the results (lines 247, 265 – 269, 271, 280 – 295, 298 – 310, 312), the discussion (lines 321 – 324, 326, 354 – 357, 378, 394 – 395, and 400), and the supplementary materials (lines 19 – 20 and 27). All figures in the main text and the supplementary materials have been updated.

Additional text has also been added to the methods section to explain and clarify the sources of current parameter estimates (lines 170 – 179) and to explain further the difficulty in obtaining transmission rate estimates for a one-health model (lines 205 – 210). The additional text reads:

“There are no direct estimates that we are aware of for the rate of emergence of resistance due to antibiotic exposure ( and ), nor for rate of emergence of resistance in the environment due to exposure to antibiotics derived from animals ( and ). Although there are promising data for estimating some transmission rates [26],[27], these are currently limited by providing information only on the number of possible transmission events between humans and animals, without environmental sampling. An estimate of the number of transmission events between humans and animals does not tell us the direction of transmission (i.e., human to animal or animal to human), nor the rate of transmission as no time period for the observed events can be provided, nor how much of this transmission was environment-mediated. Therefore, all transmission parameters in this study are estimated through fitting procedures.”

In addition we added text to the discussion section to indicate which transmission scenario would be more realistic (362 – 366).

We have made the results simpler by removing the 'baseline’ transmission scenario, and instead fitting parameters for the model without an environmental compartment for the balanced, human-, and animal-dominated scenarios. Changes to the text due to this can be found on lines 202 – 204. The supplementary table S2 was also changed to reflect this. We also simplified Figure 4 by removing the heatplot, and made Figure 3 more understandable by using point and error bars rather than a violin plot, so it is more similar to Figure 2.

Point 2: Furthermore no clear relevance between the title and the manuscript.

Response 2: We have changed the title of the manuscript from:

“The role of the environment in transmission of antibiotic resistance between humans and animals: a modelling study”

To:

“The role of the environment in transmission dynamics of antibiotic resistance in humans and animals: a modelling study”

Reviewer 3 Report

The idea and conceptualisation of the article is well defined. You conclude that there is an important role of the environment in the epidemiology of resistant bacterial infections in humans. Supplementary data about further interventions would be helpful, even if they are mentioned at the end of discussion section and based on other studies (examples of what can be done in order to reduce transmission of resistant bacteria). 

Minor spelling is required: space before citations [ ] starting from line 33 and throughout the rest of the document.

References:

- no 30 misses the italic font for the Journal name; pages and year of publication.

- no 22, 23, 24, 36, 37 need Bold font for the year of publication

Author Response

Point 1: The idea and conceptualisation of the article is well defined. You conclude that there is an important role of the environment in the epidemiology of resistant bacterial infections in humans. Supplementary data about further interventions would be helpful, even if they are mentioned at the end of discussion section and based on other studies (examples of what can be done in order to reduce transmission of resistant bacteria).

Response 1: We thank the reviewer for their encouraging comments. As suggested, we have added further examples of interventions to reduce bacterial transmission to the discussion, lines 426 – 428.

The text now reads:

“In high income countries, some studies have demonstrated transmission of resistant bacteria via the hospital environment, which can be mitigated by increased cleaning and replacement of contaminated environments like p-traps (e.g. [46],[47]).”

And the reference 46 and 47 are:

van Bunnik, B. A. D. et al (2014). Small Distances Can Keep Bacteria at Bay for Days. Proc. Natl. Acad. Sci. U. S. A. 111 (9), 3556–3560.

van der Zwet, W. C. et al (2022). Role of the Environment in Transmission of Gram-Negative Bacteria in Two Consecutive Outbreaks in a Haematology-Oncology Department. Infect. Prev. Pract. 4 (2), 100209.

Point 2: Minor spelling is required: space before citations [ ] starting from line 33 and throughout the rest of the document.

Response 2: We have added spaces before citations throughout the document.

Point 3: References:
- no 30 misses the italic font for the Journal name; pages and year of publication. - no 22, 23, 24, 36, 37 need Bold font for the year of publication

Response 3: We updated the reference section and corrected these specific issues.